# Chronotype and trait self-control as unique predictors of sleep quality in Chinese adults: The mediating effects of sleep hygiene habits and bedtime media use

**Shiang-Yi Lin**[1,2], **Kevin Kien Hoa Chung**[1,3]*

**1** Centre for Child and Family Science, The Education University of Hong Kong, Tai Po, Hong Kong SAR, China, **2** Division of Social Science, Hong Kong University of Science and Technology, Kowloon, Hong Kong SAR, China, **3** Department of Early Childhood Education, The Education University of Hong Kong, Tai Po, Hong Kong SAR, China

* kevin@eduhk.hk

**Data Availability Statement:** The data that support the findings of this study are openly available at https://doi.org/10.6084/m9.figshare.14795631.v1.

## Abstract

This study examined the distinctive roles of chronotype and trait self-control in predicting sleep quality and the mediation of sleep hygiene habits and bedtime media use of the relations between chronotype, trait self-control and sleep quality. Self-report questionnaire measuring chronotype, trait self-control, sleep hygiene behaviors, bedtime media use and sleep quality was administered to 224 Chinese adult participants (83.5% female). A multiple mediation model was estimated with sleep hygiene habits and bedtime media use as parallel mediators of the relations between chronotype, trait self-control, and sleep quality. Chronotype and trait self-control positively predicted sleep quality. Results of mediation analyses indicated that trait self-control predicted sleep quality both directly and indirectly through pre-sleep stress management and keeping a restful sleep environment, whereas chronotype predicted sleep quality indirectly through pre-sleep stress management and bedtime media use. This study provides evidence for the possible mechanism through which eveningness and low trait self-control undermine sleep quality: Whereas bedtime media use and sleep timing irregularity are linked to poor sleep quality in evening types, environmental interference (e.g., noise or disorganization) appears to be more relevant to poor sleep quality in individuals low in self-control. These findings can inform the design of personalized sleep hygiene recommendations appropriate for the target population. Practical implications regarding sleep hygiene education and interventions are discussed.

## Introduction

Poor or insufficient sleep can lead to serious consequences for physical and mental health [1]. Although much attention has been on health consequences of sleep loss, a line of sleep research has shifted the focus to behavioral factors of sleep, i.e., sleep hygiene. Sleep hygiene, for instance, having regular bedtimes or avoiding sleep-disrupting activities, has been shown to

**Funding:** This research was supported by a grant from the Positive Education in Early Childhood and Child and Family Project from the Faculty of Education and Human Development, the Education University of Hong Kong to Kevin Kien Hoa Chung.

**Competing interests:** NO authors have competing interests.

have a protective effect on sleep problems in adolescents [2]. In the present study, we proposed that two individual-difference characteristics associated with adaptive habits—i.e., chronotype and trait self-control—would predict sleep hygiene behaviors and sleep quality [3–5]. The objectives of this study were to (1) examine the relative contributions of chronotype and trait self-control to sleep quality, and (2) test sleep hygiene behaviors and bedtime media use as possible mediators of the relationships between chronotype, self-control and sleep quality. As most research on this topic tested only the mediating effect of a single pre-sleep habit (e.g., electronic media use) or a general sleep hygiene index, this study sought to provide evidence regarding whether multiple aspects of sleep hygiene behaviors, along with bedtime media use, would explain the effect(s) of chronotype and/or trait self-control on sleep quality.

## Self-control, sleep hygiene and sleep quality

Self-control entails the ability to alter, maintain or terminate one's current activity or behavior in order to meet long-term personal goals or social demands [3]. Self-control has been considered as a stable individual difference across situations and times (i.e., trait self-control) [5, 6]. Research showed that trait self-control predicted a myriad of health-related outcomes (such as healthy eating), whereas low trait self-control predicted behaviors indicative of self-regulatory problems (such as procrastination) [5, 7, 8]. Moreover, it has been suggested that trait self-control promotes sleep quality in two ways [3]. First, the capacities to enact self-control enable individuals to better organize and structure their behaviours to attain their goals (e.g., getting adequate sleep) in face of impediments or temptations that disrupt sleep (e.g., the lure of social media) [9]. Second, trait self-control underlines adaptive responses to stress and effective strategies to deal with anxiety-provoking thoughts [10].

Accumulating evidence suggests that trait self-control is associated with more adaptive habits, which enable people high in trait self-control to effortlessly engage in health-promoting behaviors or refrain from unhealthy behaviors [11]. Habits are defined as behavioral tendencies enacted with little deliberation or conscious awareness [12]. As habits are formed through repeated reward learning, habit responses can be automatically triggered in the presence of specific cues albeit no explicit goals or intentions at play [12, 13]. Although habits have been often portrayed as a challenge to self-control in pursuit of desired outcomes, a growing body of research has found that individuals high in trait self-control develop habits as a means to regulate their behavior for attaining desired outcomes. Critically, people high in trait self-control rely on beneficial habits to inhibit maladaptive impulses and initiate healthy behaviors [14, 15]. For instance, studies found that people high (vs. low) in trait self-control developed stronger habits for exercise, eating healthy snacks, and getting consistent sleep, each of which in turn predicted less efforts to engage in these healthy behaviors [15].

In sleep contexts, sleep hygiene serves as a set of adaptive behaviors that individuals can purposefully employ in order to promote positive sleep outcomes. Empirical evidence supported that higher self-control capacities (either assessed with a hand-grip task or a self-report measure) predicted better sleep hygiene [16]. Sleep hygiene comprises a wide range of behaviors that ensure sufficient and good quality of sleep, including but not limited to, maintaining regular bedtimes and rise times, reducing worry or arousing emotion before bedtime, making sleep environments comfortable, avoiding excessive exercise, and limiting caffeine, alcohol or nicotine consumption [17]. Studies reported that training of self-regulatory skills improved sleep hygiene and sleep outcomes [18]. In general, sleep hygiene accounted for 23% of the variance in subjective sleep quality in college students [19]. Pre-sleep stress management, among other sleep hygiene habits, is particularly crucial for sleep quality, given the well-documented association between stress and disturbed sleep [20]. Findings from a diary study indicated that

people high in trait self-control were less reactive and had more adaptive responses to stress [10]. Thus, we expected that sleep hygiene habits, in particular pre-sleep stress management, would mediate between trait self-control and sleep quality.

Bedtime media use has been widely studied for its negative impacts on sleep quality [21]. A growing body of research has linked low trait self-control to procrastinatory media use at bedtime [22]. For example, studies conducted among Belgian or Dutch adults found that low trait self-control predicted greater bedtime procrastination (i.e., voluntary sleep delays without obvious reasons [23]), directly and indirectly through increased deficient self-regulation via TV watching [9]. Moreover, procrastinatory TV watching was found to predict poor sleep quality via increased stress and pre-sleep cognitive arousals [24]. Given these findings, we hypothesized that people low in trait self-control (vs. those high in trait self-control) would be more likely to turn to media for instant gratification and pleasurable experience, which further undermine sleep quality [22]. We expected that bedtime media use would mediate the effect of trait self-control on sleep quality.

## Chronotype, sleep hygiene, and sleep quality

Apart from trait self-control, another individual difference associated with adaptive habits that we hypothesized would determine the ability to regulate pre-sleep behaviors is chronotype [4, 25]. Chronotype refers to an individual difference in circadian preference for sleep and activity, often mapped into the continuum of morningness-eveningness [26]. Past research has linked morningness to better sleep quality and identified eveningness as a risk factor for sleep problems, such as insufficient sleep, irregular sleep patterns, and poor sleep quality [4]. Individuals who prefer later than average sleep-wake times (i.e., evening types) are prone to experience circadian rhythm disruption due to social jetlag, operationalized as the misalignment between their preferred sleep-wake times and normative social schedules [27].

Several studies have implicated that social jetlag and sleep loss create regulatory disadvantages in daytime functioning in evening types [27–29]. Recent work using go/no-go tasks to assess sustained attention and inhibitory control in healthy young adults found that evening types had lower rate of successful inhibition [30], and that greater social jetlag was associated with poorer cognitive performance across multiple domains, particularly more impulsive response styles (i.e., fast but non-discriminatory responses) and responses that reflect inattention (i.e., more omission errors) [31]. Neuroimaging evidence confirms that social jetlag causes decreased activation in medial prefrontal cortex and dopaminergic dysfunction, the neural circuits associated with goal-directed behaviors, target detection and attention [31, 32].

We argued that social jetlag and associated problems (primarily sleep loss) may make fulfilling good sleep hygiene routines particularly challenging for evening types. On one hand, circadian misalignment causes difficulties in falling asleep in evening types when they intend to catch up normative schedules, increasing their vulnerability to developing improper sleep hygiene behaviors (such as bedtime media use). On the other hand, as has been theorized earlier, cognitive processes are vulnerable to circadian rhythm disruption [30, 31], which may continuously put demands on self-regulation among evening types and compromise their realization of proper sleep hygiene routines and inhibition of improper habits [16, 33]. For instance, a longitudinal study found a bidirectional relationship between eveningness and alcohol use in American adolescents, suggesting that circadian misalignment and maladaptive sleep hygiene habits (e.g., drinking to induce sleep) tend to reinforce each other [34].

Indeed, extensive work has shown that evening types in general have more unhealthy habitual behaviors (e.g., smoking, physical inactivity, alcohol use, and emotional eating) than morning types [4, 29]. Cross-sectional studies indicated that the association between eveningness

and maladaptive habits (e.g., procrastination) persists even when controlling for the levels of trait self-control. An initial study found that eveningness predicted lower self-control and more procrastination in Canadian college students [6]. A daily dairy study tracking Dutch workers' sleep quality and work performance for consecutive five days found that social jetlag and sleep quality jointly predicted procrastination at work: i.e., workers with greater social jet-lag also procrastinated more when they poorly slept the night before [35]. Two recent studies conducted among Polish college students also found a positive association between evening-ness and bedtime procrastination after controlling for trait self-control or general self-regula-tory skills [36, 37].

Media use has been represented as a habitualized form of behaviors [38]. Cross-sectional study findings indicated that evening types reported greater procrastinatory media use and poorer subjective sleep quality than morning types [24]. Overall, previous studies suggested bedtime media use as a potential mediator between chronotype and sleep quality. Another established link concerns the relationship between chronotype and mental health. A meta-ana-lytic report examining the link between chronotype and mental health outcomes indicated that eveningness was associated with maladaptive emotion regulation and higher levels of anx-iety and depression [4]. A study conducted among Chinese college students also found that maladaptive beliefs (such as worry) mediated between chronotype and sleep quality [39]. Thus, it appears that pre-sleep stress management would potentially mediate between chrono-type and sleep quality.

Taken together, we argued that problems in developing adaptive pre-sleep habits in evening types mainly emerge from chronic circadian disturbance and sleep deficiency. Based on the lit-erature reviewed above, we tested sleep hygiene habits as mediators that accounted for the dif-ference in sleep quality between evening and morning types. We hypothesized that pre-sleep stress management, sleep timing regularity, avoiding sleep-disrupting food or activities (such as substance use), and bedtime media use would mediate the relationship between chronotype and sleep quality.

Although chronotype and trait self-control have been well-documented as determining fac-tors for sleep behaviors, to our knowledge, no research has simultaneously examined different aspects of sleep hygiene behaviors as parallel mediators between chronotype, trait self-control, and sleep quality. In the present study, we took a multiple mediation approach where the indi-rect effects associated with each pre-sleep habit were simultaneously tested for their mediating roles in the relations between chronotype, trait self-control and sleep quality. Assessing multi-ple mediation was to not only identify whether a mediation effect exists, but also determine the shared and unique contributions of several pre-sleep habits that overlap in content [40]. Moreover, because a multiple mediation model evaluates the indirect effect of each mediator in the presence of the other mediators, it substantially reduces the risk of parameter bias due to omitted variables in single mediation models [40]. In the present study, we expected that indi-vidual differences in trait self-control and chronotype contributed to different sleep hygiene behaviors, which in turn predicted subjective sleep quality. We hypothesized that morningness and higher trait self-control would be associated with more adaptive sleep hygiene habits and better sleep quality, and that sleep hygiene habits and bedtime media use would mediate the associations between chronotype, trait self-control and sleep quality.

## Methods

### Participants and procedure

The present study draws on data from a cross-sectional study on Chinese adults' sleep behavior and sleep quality. Participants were recruited online by convenient sampling. Inclusion criteria

was age 20 or older, proficient in Chinese languages, and currently residing in Hong Kong. Data were collected between 21 December 2020 and 15 January 2021. Every participant took part in the study on a voluntary basis. Participants completed the web-based survey administered in Chinese in approximately 25–35 minutes and received a supermarket coupon of $25 HKD (= $3.22 USD) for their participation. Ethics approval was from the Human Research Ethnics Committee of the Education University of Hong Kong. Written informed consent from each participant was obtained before the survey. To ensure data quality, three attention checks were embedded throughout the study: in each question, participants were instructed to choose a specific answer, or not to choose any answer at all. Sixteen participants who failed all the checks were excluded from the analyses. Additionally, 8 participants who reported to be shift workers were also excluded. The final sample consists of 224 adult participants, predominantly female (83.5%), aged below 40 years old (75.4%).

## Measures

The questionnaire consisted of the adult sleep-awake scale, the sleep hygiene index, the brief self-control scale, a measure of bedtime media use, the reduced Morningness-Eveningness Questionnaire (described in more details below) and demographic information, including age, gender, and marital status. The English to Chinese translation was done by the first author. Back translation was done by two research assistants. The Chinese translation was revised and polished with careful examination of the back translations by other members of the research team.

**Sleep quality.** Sleep quality was assessed using the Adult Sleep-Awake Scale [41]. Example items are "After waking up during the night, I roll over and go right back to sleep" and "After I fall asleep, but during the night, I toss and turn in bed" (reverse coded). The scale contains sleep behaviors on five dimensions—going to bed, falling asleep, maintaining sleep, reinitiating sleep, and returning to wakefulness. Participants rated each item on a 6-point scale from 1 "never" to 6 "always" with respect to how frequently each behavior occurred. The scale showed good internal consistency ($\alpha$ = .90). A higher mean score indicates better sleep quality.

**Bedtime media use.** Bedtime media use was measured with a set of questions developed by the authors to assess the frequency and duration of bedtime media use from Monday to Thursday night in the past month. Three items assessing the frequency of bedtime media use include "How often do you use social media (e.g., Facebook, Instagram or WhatsApp) for texting, surfing, and other purposes with an electronic device, including a laptop, tablet, or mobile phone right before going to sleep?"; "How often do you watch videos/films/tv series, or listen to music (e.g., Netflix or YouTube) with an electronic device right before going to sleep?"; "How often do you play games with an electronic device including a mobile phone or gaming console right before going to sleep?" (4-point scale from 1 "never" to 4 "almost everyday"). Two items assessed the duration of sleep delays due to media use, which read "For how many minutes do you think that you put off sleep due to social media use at bedtime?" and "For how many minutes do you think that you put off sleep to watch videos/films/tv series, listen to music, or play games at bedtime?" Following Fossum et al.'s (2014) suggestion, the total time of sleep delays due to bedtime media use was then calculated by multiplying the frequency (the number of days) by duration (the minutes of bedtime delays on average) for each type of media [42]. The total sleep delay was then standardized, with a higher score indicating greater bedtime media use.

**Chronotype.** Chronotype was measured using the reduced Morningness-Eveningness Questionnaire (rMEQ) [43]. The five items assess participants' typical rise time, morning freshness, typical retire time, subjective peak time ("At what time of the day do you think that

you reach your feeling best peak?") and their self-evaluation of chronotype ("One hears about morning and evening types of people. Which one of these types do you consider yourself to be?") on a 4-point scale from 1 "definitely an evening type" to 4 "definitely a morning type." The total scores were computed following the guideline by Adan and Almirall [43]. The total scores ranged from 4 to 26, with higher scores indicating greater morningness preferences. The internal consistency of the rMEQ in the present study was .61, similar to the scale reliability reported in past research [4, 44].

**Sleep hygiene behaviors.** The 13-item Sleep Hygiene Index (SHI) [45] was used to assess sleep hygiene behaviors over the past month (e.g., "I get out of bed at different times from day to day"; "I go to bed feeling stressed, angry, upset, or nervous"). Participants rated how often they have engaged in each behavior over the past month on a 5-point scale ranging from 1 "never" to 5 "always." The scores of all the items were reverse coded, with a higher score indicating better sleep hygiene. An exploratory factor analysis was performed using the principal component method and Oblimin rotation. Kaiser-Meyer-Olkin (KMO) measure and the Bartlett's test of Sphericity were used to detect sampling adequacy and data distribution, respectively. The KMO value was .77 and the Bartlett test reached significance ($p < .001$), indicating the suitability and adequacy of the sample for factor analysis.

A four-factor solution emerged. The factors cumulatively accounted for 60.6% of the total variance of item scores (Factor 1 = 27.9%, Factor 2 = 12.9%, Factor 3 = 11.5%, Factor 4 = 8.2%). Factors 1–4 are referred to as "pre-sleep stress management" ($\alpha$ = .72); "sleep timing regularity" ($\alpha$ = .63); "avoiding sleep-disrupting food or activities" ($\alpha$ = .60); "keeping a restful sleep environment" ($\alpha$ = .72), respectively (see Table 1 for the items and factor loadings). A confirmatory factor analysis (CFA) was then conducted to verify the factor structure. The fit indices used to assess model fit were the comparative fit index (CFI), Tucker-Lewis Index (TLI), the root-mean-square error of approximation (RMSEA) and standardized root mean square residual (SRMR). Values of CFI and TLI > .90, RMSEA < .06, and SRMR < .08 indicate a good fit with the data [46]. The results of CFA indicated that the four-factor model provided a reasonable fit for the data, CFI = .94, TLI = .91, RMSEA = .06, SRMR = .06. The mean scores of the items loaded on each factor were computed for mediation analysis.

**Table 1. Items and factor loadings of the Sleep Hygiene Index (SHI).**

| Item Content | Factor Loadings | | | |
|---|---|---|---|---|
| | Factor 1 | Factor 2 | Factor 3 | Factor 4 |
| 1. I take daytime naps lasting two or more hours | | | .56 | |
| 2. I go to bed at different times from day to day | | .46 | | |
| 3. I get out of bed at different times from day to day | | .40 | | |
| 4. I exercise to the point of sweating within 1 h of going to bed | | | .31 | |
| 5. I stay in bed longer than I should two or three times a week | | .78 | | |
| 6. I use alcohol, tobacco, or caffeine within 4 h of going to bed or after going to bed | | | .44 | |
| 7. I do something that may wake me up before bedtime | | .43 | | |
| 8. I go to bed feeling stressed, angry, upset, or nervous | .68 | | | |
| 9. I use my bed for things other than sleeping | .68 | | | |
| 10. I sleep on an uncomfortable bed | | | | .66 |
| 11. I sleep in an uncomfortable bedroom | | | | .70 |
| 12. I do important work before bedtime | .65 | | | |
| 13. I think, plan, or worry when I am in bed | .70 | | | |

All items were reverse coded, with higher scores indicating better sleep hygiene.

**Trait self-control.** Trait self-control was measured with the 13-item brief Self-Control Scale [6], which assesses perceptions of self-discipline, impulsivity, and resistance to temptation over the past month. Example items are "I am able to work effectively toward long-term goals" and "I wish I had more self-discipline" (reverse coded). The items were rated on a 5-point scale from 1 "not descriptive at all" to 5 "very much descriptive." The scale showed good internal consistency ($\alpha$ = .83). A higher mean score indicates higher trait self-control.

**Control variables.** Participants' demographic characteristics (i.e., age, gender, and marital status) served as control variables. Participants indicated their age group (21–30 years, 31–40 years, 41–50 years, 51–60 years, or 60 years or above), gender (male vs. female) and marital status (single, married, divorced or widowed). As age was assessed as a categorical variable, two dummy-coded variables were created (21–30 years vs. 31–50 years, and 21–30 years vs. 51 years or above). Marital status was recoded into one dummy-coded variable (married vs. single, widowed, or divorced).

## Analytic plan

Data analyses were conducted using IBM SPSS 26. Preliminary analysis was first undertaken to examine demographic characteristics of the total sample, the distributions of chronotype and trait self-control, and the frequency and duration of bedtime media use. To provide an overview of chronotype in relation to pre-sleep habits, we examined the differences in sleep hygiene habits and sleep behaviors between circadian typology groups classified based on Adan and Almirall's (1991) criterion (evening types < 12; neither: 12–17; morning types > 17) [43]. For correlational and mediation analyses, chronotype (morningness-eveningness) and trait self-control were used as continuous scores. Prior to mediation analyses, Pearson's $r$ was used to assess the bivariate correlations between the study variables, whereas Spearman's $\rho$ was computed to examine the associations between demographic characteristics (age, gender, and marital status) and study variables. To test the proposed pathways, mediation analyses were carried out using PROCESS macro for SPSS [47] to examine whether chronotype and trait self-control were associated with sleep quality and whether the relationships were mediated by sleep hygiene habits and bedtime media use. Chronotype and trait self-control were entered as independent variables to predict sleep quality, and sleep hygiene habits and bedtime media use were entered as parallel mediators (see Fig 1 for the proposed multiple mediation model). Participant age, gender, and marital status were included as covariates in the mediation analyses. Indirect effects of mediating effects were calculated using 5,000 bootstrap samples with 95% confidence intervals (CIs), whereby a significant mediating effect was determined by the exclusion of 0.

## Results

### Preliminary analyses

**Demographics.** Table 2 depicts the descriptive statistics of demographic characteristics. A total of 224 Chinese adults completed the survey (83.5% female). The majority of participants aged 40 or below (38.8% 20–30 years old, 36.6% 31–40 years old), with another 16.5% and 8% reporting to be 41–50 years old and 51 years old or above, respectively. About half of participants were single (50.9%), 46.0% were married, and 3.1% were divorced or widowed.

**Descriptive statistics for chronotype and trait self-control.** Table 2 presents descriptive statistics for all the study variables. In the total sample, the mean for chronotype (rMEQ) was 13.22 (*SD* = 3.36; range = 5–26), and the mean for trait self-control was 3.09 (*SD* = .59, range = 1–4). Both the rMEQ scores (skewness: .15; kurtosis: -.28) and trait self-control scores (skewness: .43; kurtosis: .05) were normally distributed (see Fig 2).

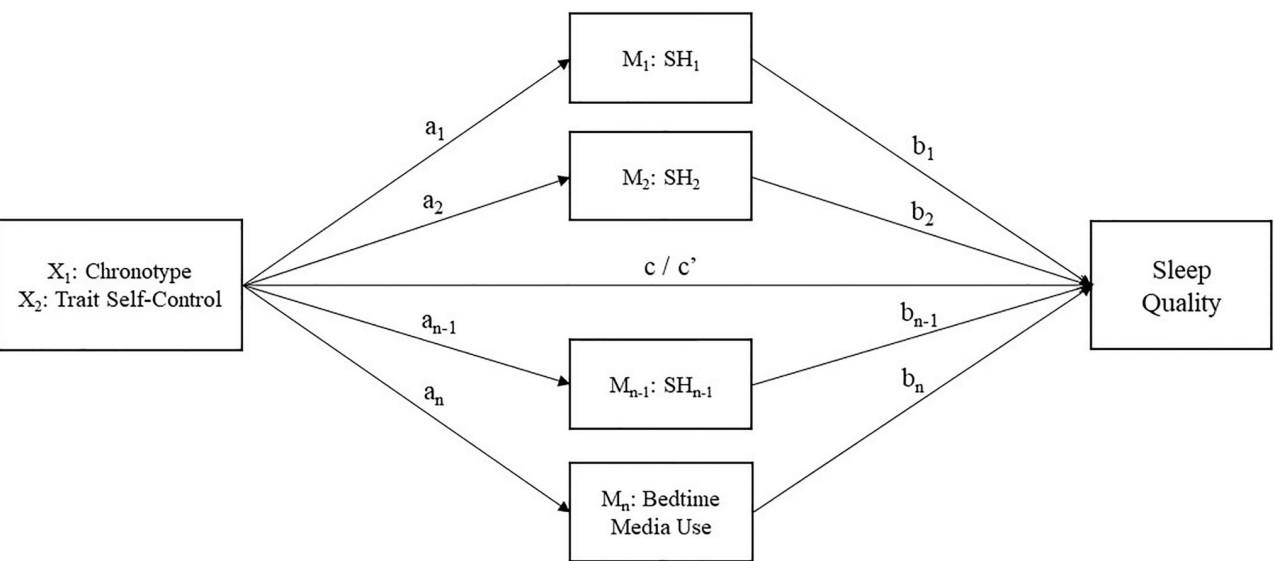

**Fig 1. Proposed multiple mediation model.** Chronotype and trait self-control were considered as independent variables ($X_1$, $X_2$), sleep hygiene habits ($M_1$–$M_{n-1}$) and bedtime media use ($M_n$) as mediators and sleep quality as outcome variable (Y). X can affect Y either directly (c') or indirectly through $M_1$-$M_n$ ($a_1$-$a_n$, $b_1$-$b_n$ paths). Age, gender, and marital status were included as covariates. N represents the number of mediators, and n-1 was determined by the number of sleep hygiene components derived from exploratory factor analysis of the item scores of Sleep Hygiene Index.

The distribution of circadian typology groups was 69 participants (30.8%) being evening types, 131 (58.5%) being intermediate types, and 24 (10.7%) being morning types. The distribution of circadian typology groups was consistent with that observed in past research using rMEQ in Western adult populations, in which the most frequent group was intermediate type [43]. Looking at the demographic characteristics of circadian typology groups, although evening types appeared to consist of a higher percentage of young, single adults than intermediate or morning types, the tests of significance found only the difference in marital status across circadian typology groups, $\chi^2(6) = 19.40$, $p = .004$, whereas gender and age differences were non-significant, $\chi^2(2) = 1.14$, $p = 57$; $\chi^2(8) = 11.93$, $p = .15$. With regard to pre-sleep habits and sleep behaviors, the morning group appeared to score higher on subjective sleep quality, morningness preference, trait self-control, good sleep hygiene (except avoiding sleep-disrupting food or activities), as well as reported to spend less time on bedtime media use than intermediate or morning types.

**Frequency and duration of bedtime media use.** Table 3 presents the frequency and duration of bedtime media use by media type. The majority (80.4%) of the participants reported to engage in social media surfing/texting before going to sleep almost every day over the past month. Whereas 58.9% of the participants reported to watch films, videos or TV series, or listen to music at least 3 times per week, 21.9% of them reported to play games on electronic devices at bedtime at least 3 times per week. With respect to the average duration of media-related sleep delays, participants reported to put off their sleep due to social media use for 39.70 minutes on average (SD = 31.94 minutes) and due to other media activities (i.e., TV watching, listening to music, or gaming) for 40.01 minutes on average (SD = 44.34 minutes).

**Bivariate associations.** Table 4 presents bivariate correlations between the study variables. Sleep quality, as predicted, was positively correlated to chronotype, self-control, all the sleep hygiene behaviors (rs = .29~.57, ps < .0001), but negatively correlated with bedtime media use (r = -.36, p < .0001). Chronotype was weakly and positively associated with self-control (r =

**Table 2. Descriptive statistics of the total sample and by circadian typology.**

| Measures/Categories | | Total sample | Circadian typology | | |
|---|---|---|---|---|---|
| | | (N = 224) | Evening (N = 69) | Intermediate (N = 131) | Morning (N = 24) |
| | | Percentage | Percentage | Percentage | Percentage |
| Age | | | | | |
| 21–30 | | 38.8% | 52.2% | 35.1% | 20.8% |
| 31–40 | | 36.6% | 29.0% | 38.2% | 50.0% |
| 41–50 | | 16.5% | 14.5% | 17.6% | 16.7% |
| 51–60 | | 4.9% | 2.9% | 6.1% | 4.2% |
| 60 or above | | 3.1% | 1.4% | 3.1% | 8.3% |
| Gender | | | | | |
| Male | | 16.5% | 20.3% | 15.3% | 12.5% |
| Female | | 83.5% | 79.7% | 84.7% | 87.5% |
| Marital status | | | | | |
| Single | | 50.9% | 63.8% | 47.3% | 33.3% |
| Married | | 46.0% | 33.3% | 51.1% | 54.2% |
| Divorced | | 2.7% | 2.9% | 1.5% | 8.3% |
| Widowed | | .4% | - | - | 4.2% |
| | Range | Mean (SD) | Mean (SD) | Mean (SD) | Mean (SD) |
| Sleep quality | 1–6 | 4.24 (.69) | 4.1 (.72) | 4.25 (.60) | 4.64 (.90) |
| Chronotype (rMEQ) | 5–26 | 13.22 (3.36) | 9.45 (1.60) | 14.12 (1.71) | 19.17 (1.27) |
| Trait self-control | 1–4 | 3.09 (.59) | 2.93 (.54) | 3.12 (.57) | 3.37 (.73) |
| SH1 | 1–5 | 3.32 (.84) | 3.16 (.80) | 3.31 (.84) | 3.81 (.76) |
| SH2 | 1–5 | 3.26 (.75) | 2.98 (.73) | 3.34 (.71) | 3.64 (.77) |
| SH3 | 1–5 | 4.45 (.56) | 4.37 (.60) | 4.49 (.55) | 4.43 (.50) |
| SH4 | 1–5 | 4.08 (.81) | 4.05 (.85) | 4.03 (.80) | 4.44 (.60) |
| Bedtime media use [a] | | 274 (221) | 321 (249) | 243 (208) | 157 (190) |

Note. SH1 = pre-sleep stress management, SH2 = sleep timing regularity, SH3 = avoiding sleep-disrupting food or activities, SH4 = keeping a restful sleep environment. rMEQ = the reduced Morningness-Eveningness Questionnaire.

[a] Bedtime media use refers to the total minutes of sleep delays due to media use at bedtime per week.

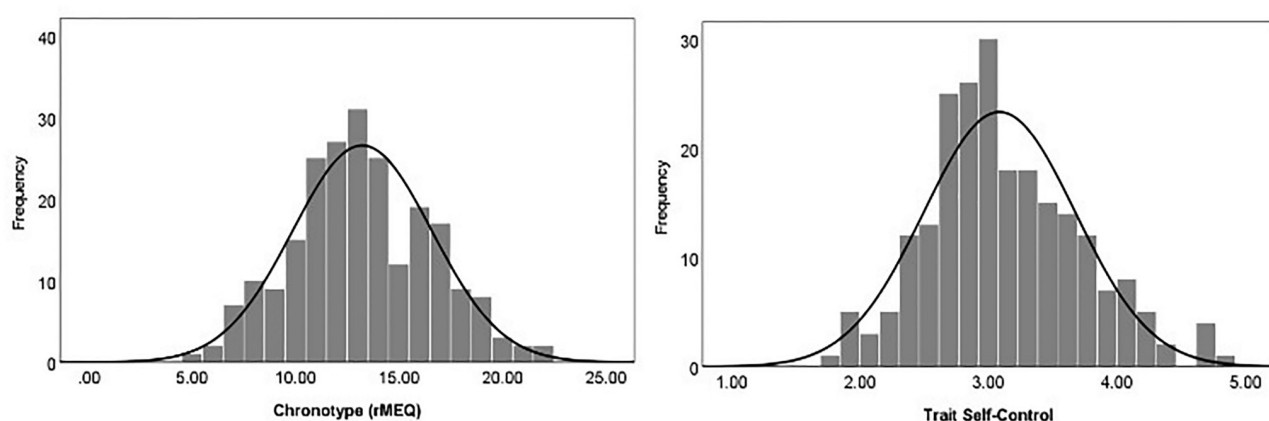

**Fig 2. Distributions of chronotype (rMEQ) and trait self-control in the total sample.**

**Table 3. Frequency and duration of bedtime media use by media type.**

| Media Type | N | Frequency (per week) | | | | Duration (minutes per night) | |
|---|---|---|---|---|---|---|---|
| | | **Never** | **1–2 times** | **3–5 times** | **Almost everyday** | *Mean* | *SD* |
| Social media surfing/texting (e.g., Facebook, Instagram, or WhatsApp) | 224 | 2.2% | 7.1% | 10.3% | 80.4% | 39.70 | 31.94 |
| Watching films/videos/TV series or listening to music (e.g., Netflix or YouTube) | 224 | 17.4% | 23.7% | 21.9% | 37.0% | 40.01 | 44.34 |
| Playing games on any electronic device | 224 | 56.2% | 21.9% | 9.4% | 12.5% | | |

Note. *SD* = standard deviation.

.22, $p$ = .001). Furthermore, chronotype was positively correlated with pre-sleep stress management ($r$ = .27, $p$ < .0001) and sleep time regularity ($r$ = .37, $p$ < .0001), but was negatively correlated with bedtime media use ($r$ = -.28, $p$ < .001). On the other hand, self-control was weakly and positively correlated with all the sleep hygiene behaviors ($rs$ = .16~.22, $ps$ < .02), but was negatively correlated with bedtime media use ($r$ = -.15, $p$ = .03). The results of the associations between demographic characteristics and study variables indicated that as compared to younger adults (20–30 years old), the mid-age group (31–50 years) had greater morningness preference ($\rho$ = .16, $p$ = .02), and older adults (51 years or above) had better pre-sleep stress management ($\rho$ = .22, $p$ = .001), sleep time regularity ($\rho$ = .17, $p$ = .01), and lesser bedtime media use ($\rho$ = -.20, $p$ = .002). Being married (vs. single) was positively correlated with morningness ($\rho$ = .21, $p$ = .002), self-control ($\rho$ = .18, $p$ = .008), pre-sleep stress management ($\rho$ = .22, $p$ = .001) and sleep time regularity ($\rho$ = .17, $p$ = .01). None of the demographic variables were correlated with sleep quality ($\rho s$ < .08, $ps$ > .21).

**Table 4. Bivariate associations between study variables.**

| | | 1 | 2 | 3 | 4 | 5 | 6 | 7 | 8 | 9 | 10 | 11 | 12 |
|---|---|---|---|---|---|---|---|---|---|---|---|---|---|
| 1 | Age D1 | - | -.31** | .06 | .36** | .08 | .16* | .13 | .08 | .03 | .02 | .06 | .06 |
| 2 | Age D2 | | - | .04 | .25** | .06 | .11 | .13 | .22** | .17* | .01 | -.07 | -.20** |
| 3 | Gender | | | - | .07 | .04 | .07 | .06 | -.07 | .01 | .09 | -.07 | .01 |
| 4 | Marital status | | | | - | .08 | .21** | .18** | .22** | .17* | .05 | -.01 | -.02 |
| 5 | Sleep quality | | | | | - | .30** | .36** | .57** | .38** | .29** | .46** | -.36** |
| 6 | Chronotype (Morningness) | | | | | | - | .22** | .27** | .37** | .05 | .12 | -.28** |
| 7 | Trait self-control | | | | | | | - | .20** | .22** | .16* | .16* | -.15* |
| 8 | SH1 | | | | | | | | - | .42** | .23** | .33** | -.29** |
| 9 | SH2 | | | | | | | | | - | .28** | .20** | -.26** |
| 10 | SH3 | | | | | | | | | | - | .33** | -.15* |
| 11 | SH4 | | | | | | | | | | | - | -.26** |
| 12 | Bedtime media use | | | | | | | | | | | | - |

Note. Age D1 was coded as 0 for 21–30 years and 1 for 31–50 years, whereas age D2 was coded as 0 for 21–30 years and 1 for 50 years or above. Gender was coded 0 for men and 1 for women. Marital status was coded as 0 for single, divorced or widowed and 1 for married. SH1 = pre-sleep stress management, SH2 = sleep timing regularity, SH3 = avoiding sleep-disrupting food or activities, SH4 = keeping a restful sleep environment.

*$p$ < .05.

**$p$ < .01.

## Mediation analyses

The total effect of trait self-control on sleep quality was significant, with higher trait self-control predicting better sleep quality ($b = .32$, $SE = .08$, $p < .0001$, 95%CI = .17 to .47). As shown in Table 5, higher trait self-control was associated with slightly better stress management ($b = .17$, $SE = .09$, $p = .065$), greater sleep timing regularity ($b = .17$, $SE = .08$, $p = .039$), and avoiding sleep-disrupting food or activities ($b = .14$, $SE = .07$, $p = .041$), and keeping a restful sleep environment ($b = .21$, $SE = .09$, $p = .024$). Nonetheless, self-control was unrelated to bedtime media use ($b = -.19$, $SE = .12$, $p = .102$). Furthermore, among the sleep hygiene behaviors, only pre-sleep stress management and keeping a restful sleep environment had significant direct effects on sleep quality ($b = .32$, $SE = .05$, $p < .001$; $b = .19$, $SE = .05$, $p < .0001$), whereas the direct effects of sleep timing regularity and avoiding sleep-disrupting food or activities on sleep quality were non-significant ($ts < .99$, $ps > .32$). Bedtime media use also significantly and negatively predicted sleep quality ($b = -.09$, $SE = .04$, $p = .029$). Table 6 presents the bootstrapping results of the total, direct and indirect effects of the multiple mediation model. Results of the bootstrapped CIs for the indirect effects of self-control on sleep quality indicated significant mediation through pre-sleep stress management ($b = .06$, $SE = .03$, 95%CI = .002 to .123) and keeping a restful sleep environment ($b = .04$, $SE = .02$, 95%CI = .005 to .089). After controlling for the mediators, the residual direct effect of self-control on sleep quality was still significant ($b = .14$, $SE = .05$, $p = .0003$, 95%CI = .104 to .347).

There was also a significant total effect of chronotype on sleep quality, with greater morningness predicting better sleep quality ($b = .05$, $SE = .01$, $p = .0002$, 95%CI = .024 to .076). As shown in Table 5, chronotype significantly predicted better pre-sleep stress management ($b = .05$, $SE = .02$, $p = .002$), greater sleep timing regularity ($b = .07$, $SE = .01$, $p < .001$), and less bedtime media use ($b = -.18$, $SE = .02$, $p < .001$), but chronotype was unrelated to avoiding sleep-disrupting food or activities and keeping a restful sleep environment ($ts < 1.44$, $ps > .15$). As shown in Table 6, the results of the bootstrapped CIs for the indirect effect of chronotype on

**Table 5. Summary of regression coefficients for the multiple mediation model.**

| Predictors | M1: SH1 | | M2: SH2 | | M3: SH3 | | M4: SH4 | | M5: Bedtime media use | | Y: Sleep quality | |
|---|---|---|---|---|---|---|---|---|---|---|---|---|
| | B (SE) | p | B (SE) | p | B (SE) | p | B (SE) | p | B (SE) | p | B (SE) | p |
| Constant | 3.34 (.14) | .000 | 3.29 (.13) | .000 | 4.35 (.1) | .000 | 4.26 (.15) | .000 | -.04 (.18) | .826 | 1.95 (.33) | .000 |
| Age D1 | .12 (.13) | .336 | -.06 (.11) | .619 | -.01 (.09) | .946 | .01 (.13) | .955 | .24 (.16) | .132 | .01 (.08) | .862 |
| Age D2 | .58 (.23) | .010 | .29 (.2) | .142 | -.04 (.16) | .794 | -.2 (.23) | .383 | -.29 (.28) | .298 | -.16 (.15) | .288 |
| Gender | -.25 (.14) | .082 | -.1 (.12) | .424 | .08 (.1) | .408 | -.18 (.14) | .210 | -.15 (.18) | .396 | .05 (.10) | .599 |
| Marital status | .15 (.12) | .220 | .12 (.11) | .278 | .08 (.09) | .361 | -.03 (.13) | .808 | .13 (.16) | .409 | -.06 (.08) | .484 |
| Trait self-control | .17 (.09) | .065 | .17 (.08) | .039 | .14 (.07) | .041 | .21 (.09) | .024 | -.19 (.12) | .102 | .19 (.06) | .002 |
| Chronotype (Morningness) | .05 (.02) | .002 | .07 (.01) | .000 | .00 (.01) | .999 | .02 (.02) | .152 | -.08 (.02) | .000 | .01 (.01) | .327 |
| M1: SH1 | | | | | | | | | | | .32 (.05) | .000 |
| M2: SH2 | | | | | | | | | | | .05 (.06) | .386 |
| M3: SH3 | | | | | | | | | | | .07 (.07) | .326 |
| M4: SH4 | | | | | | | | | | | .19 (.05) | .000 |
| M5: Bedtime media use | | | | | | | | | | | -.09 (.04) | .029 |
| | $R^2 = .16$ | | $R^2 = .18$ | | $R^2 = .03$ | | $R^2 = .05$ | | $R^2 = .13$ | | $R^2 = .47$ | |

Note. Age D1 was coded as 0 for 21–30 years and 1 for 31–50 years, whereas age D2 was coded as 0 for 21–30 years and 1 for 50 years or above. Gender was coded 0 for men and 1 for women. Marital status was coded as 0 for single, divorced or widowed and 1 for married. SH1 = pre-sleep stress management, SH2 = sleep timing regularity, SH3 = avoiding sleep-disrupting food or activities, SH4 = keeping a restful sleep environment.

**Table 6. Bootstrapping results of the total, direct and indirect effects of multiple mediation model.**

| Effects/variables | B | SE | 95%CI |
|---|---|---|---|
| Total effect of trait self-control on sleep quality | .368 | .075 | [.221, .515] |
| Direct effect of trait self-control on sleep quality | .226 | .062 | [.104, .347] |
| Total indirect effect of trait self-control on sleep quality | .142 | .049 | [.044, .237] |
| Through SH1 | .060 | .030 | [.002, .123] |
| Through SH2 | .008 | .011 | [-.011, .035] |
| Through SH3 | .013 | .011 | [-.006, .038] |
| Through SH4 | .044 | .022 | [.005, .089] |
| Through bedtime media use | .018 | .014 | [-.002, .051] |
| Total effect of chronotype on sleep quality | .050 | .013 | [.024, .076] |
| Direct effect of chronotype on sleep quality | .018 | .012 | [-.005, .041] |
| Total indirect effect of chronotype on sleep quality | .032 | .008 | [.015, .047] |
| Through SH1 | .016 | .005 | [.007, .027] |
| Through SH2 | .003 | .004 | [-.005, .011] |
| Through SH3 | .000 | .001 | [-.002, .003] |
| Through SH4 | .005 | .004 | [-.001, .013] |
| Through bedtime media use | .007 | .003 | [.001, .014] |

Note. *B* = unstandardized coefficient, CI = confidence interval, *SE* = standard error. SH1 = pre-sleep stress management, SH2 = sleep timing regularity, SH3 = avoiding sleep-disrupting food or activities, SH4 = keeping a restful sleep environment.

sleep quality indicated significant mediation through pre-sleep stress management ($b = .02$, *SE* = .005, 95%CI = .007 to .027) and bedtime media use ($b = .007$, *SE* = .003, 95%CI = .001 to .014). After controlling for the mediators, the residual direct effect of chronotype on sleep quality was no longer significant ($b = .02$, *SE* = .01, $p = .12$, 95%CI = -.005 to .041). The mediation effects remained unchanged when control variables were omitted from the model.

## Additional analysis: Moderated mediation

We ran additional analyses using PROCESS macro for SPSS to examine whether chronotype or trait self-control moderated significant direct and indirect paths in the multiple mediation model reported earlier (i.e., moderated mediation). First, we found that chronotype and trait self-control jointly predicted pre-sleep stress management ($b = .06$, *SE* = .02, $p = .021$) and sleep timing regularity ($b = .05$, *SE* = .02, $p = .030$; see S1 Table). Results of simple main effect analysis revealed that higher trait self-control predicted more adaptive sleep hygiene habits, in particular pre-sleep stress management and sleep timing regularity, only among morning types (pre-sleep stress management: $b = .34$, *SE* = .12, 95%CI = .117 to .572; sleep timing regularity: $b = .32$, *SE* = .10, 95%CI = .115 to .517), but not among evening types (pre-sleep stress management: $b = -.06$, *SE* = .13, 95%CI = -.324 to .198; sleep timing regularity: $b = -.03$, *SE* = .12, 95%CI = -.260 to .200; see Fig 3).

Second, chronotype moderated the direct path between pre-sleep stress management and sleep quality ($b = -.04$, *SE* = .02, $p = .016$; See S1 Table). Specifically, the effect of pre-sleep stress management on sleep quality was stronger for evening types ($b = .42$, *SE* = .07, 95%CI = .282 to .563) than morning types ($b = .16$, *SE* = .08, 95%CI = .013 to .308), suggesting that pre-sleep stress management was particularly important for improving sleep quality in evening types, in that evening types who established better stress management habits reported equally good sleep quality as morning types (see Fig 4).

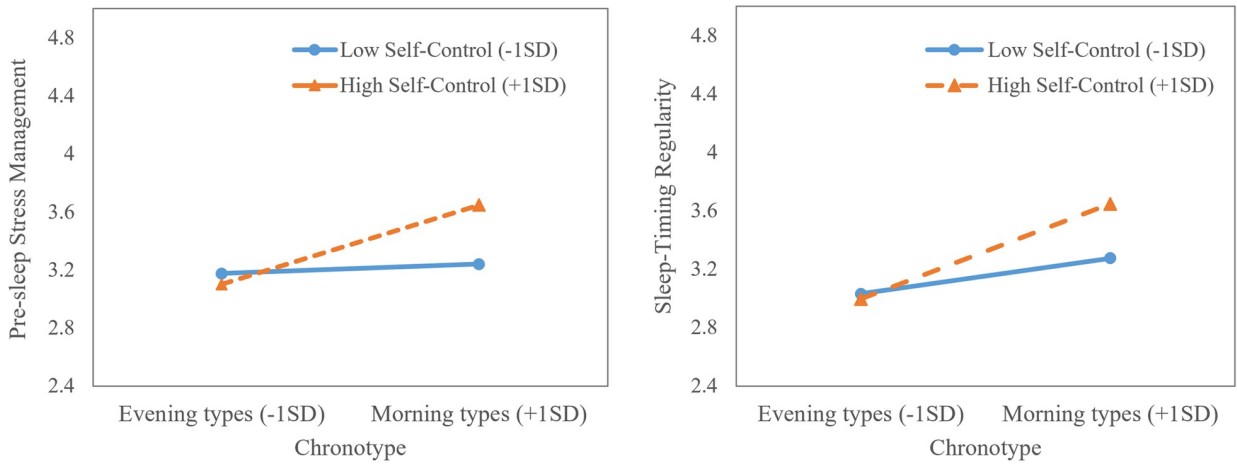

**Fig 3. Pre-sleep stress management and sleep timing regularity as a function of chronotype and trait self-control.** The interactions were probed at high (1SD above the mean) and low levels (1SD below the mean) of chronotype and trait self-control.

Finally, we found that the indirect effect of chronotype to sleep quality through pre-sleep stress management was moderated by trait self-control: For individuals high in trait self-control, more pronounced morningness predicted better pre-sleep stress management, which in turn predicted better sleep quality ($b = .03$, $SE = .01$, 95%CI = .013 to .048). The indirect effect

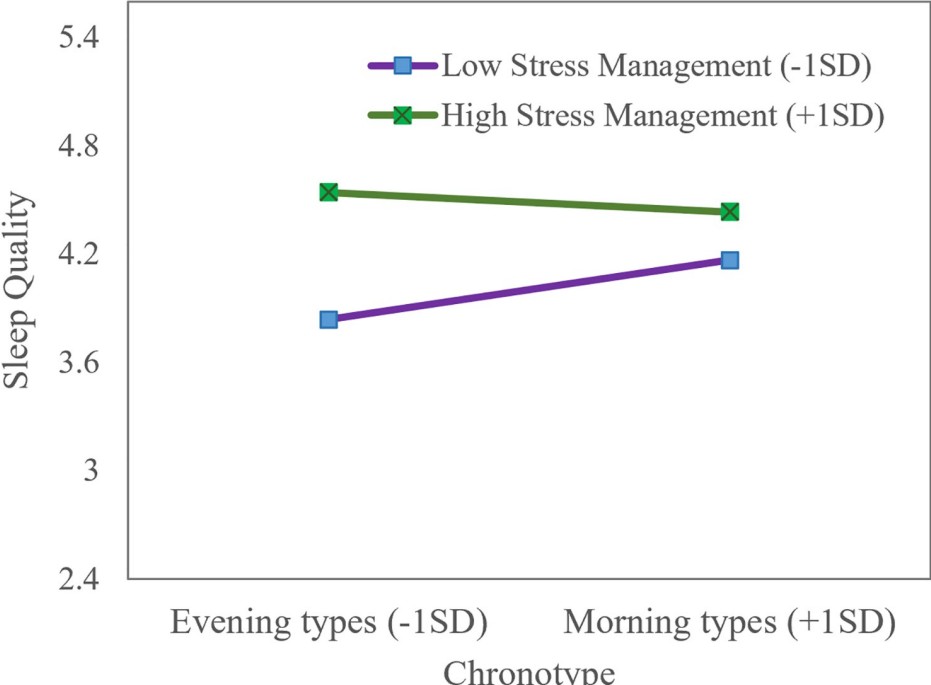

**Fig 4. Sleep quality as a function of chronotype and levels of pre-sleep stress management.** The interactions were probed at high (1SD above the mean) and low levels (1SD below the mean) of chronotype and pre-sleep stress management.

was diminished and non-significant for those low in trait self-control ($b = .00$, $SE = .01$, 95%CI = -.008 to .014). The results of additional analyses further supported that chronotype and trait self-control make unique, separate contributions to sleep hygiene behaviors and sleep quality. Importantly, the effects of trait self-control and chronotype on sleep quality are likely to be interdependent, such that the presence of either risk factor (low trait self-control or eveningi-ness) would undermine sleep quality.

## Discussion

The main objectives of this study were to examine the distinctive roles of chronotype and trait self-control in predicting sleep quality and the mediating effects of sleep hygiene behaviors and bedtime media use on the associations between chronotype, trait self-control and sleep quality in Chinese adults in Hong Kong. The results showed that morningness and higher trait self-control were related to better sleep quality. The multiple mediation model revealed that pre-sleep stress management appeared to be the most significant mediator of the relationships between chronotype, trait self-control and sleep quality even in the presence of the other medi-ators. The multiple mediation analyses further pinpointed the unique contributions of chrono-type and trait self-control to sleep quality through different mediation processes: Whereas chronotype predicted sleep quality indirectly via pre-sleep stress management and bedtime media use, trait self-control predicted sleep quality both directly and indirectly via pre-sleep stress management and keeping a restful sleep environment. The significant residual direct effect of trait self-control on sleep quality was consistent with past research findings about the presence of both direct and indirect effects of trait self-control on health behavior of other domains [48].

Findings concerning the associations between chronotype, self-control and sleep quality echoed past research that has identified eveningness and low self-control as risk factors for poor and insufficient sleep [4, 5, 29]. The finding that pre-sleep stress management mediated between both trait self-control and sleep quality, and between chronotype and sleep quality indicates that morning types or people high in self-control had better pre-sleep stress manage-ment, which in turn promoted sleep quality. The results are largely consistent with research findings that a lack of stress management skills partly explained poor sleep quality in evening types or people low in trait self-control [20, 39], suggesting that pre-sleep stress management, including reducing worry and other arousing emotion before bedtime, is crucial for alleviating sleep problems. In other words, adaptive pre-sleep habits that foster self-regulation abilities to avoid engaging in emotion-provoking activities underlie the differences in sleep quality between morning and evening types, and between people high vs. low in trait self-control [3, 16].

The absence of the mediation effects of sleep timing regularity and avoiding sleep-disrupt-ing food or activities indicates that the indirect effects associated with the two behaviors were largely reduced in the presence of the other mediators. To account for this possibility, we rees-timated the mediation model by removing the effects of pre-sleep stress management and keeping a restful sleep environment. We found a significant indirect effect of sleep timing reg-ularity between chronotype and sleep quality ($b = .05$, $SE = .02$, 95%CI = .004 to .097) and a significant indirect effect of avoiding sleep-disrupting food or activities between trait self-con-trol and sleep quality ($b = .03$, $SE = .02$, 95%CI = .003 to .063). This attenuation effect suggests pre-sleep stress management and keeping a restful sleep environment are the pervading deter-minants of sleep quality in the current study context.

Bedtime media use, defined as sleep delay due to media use at bedtime, was found to medi-ate the link between chronotype and sleep quality, but not between self-control and sleep

quality. In line with research that has linked eveningness to bedtime procrastination [25, 36, 37]. This finding suggests that evening types were more inclined to spend time on media and put off their bedtime than morning types. It is important to note that the findings did not rule out the possibility that bedtime media use also exacerbates disruption of circadian rhythm and causes sleep delay in evening types via other mechanisms. For instance, blue light emitted from electronic devices can inhibit melatonin production, and media activities increase cognitive and emotional arousals [27].

An unanticipated result of the present study was that bedtime media use did not mediate between trait self-control and sleep quality. This hypothesis was built upon a large body of research literature indicating that bedtime procrastination stems from self-regulatory failures [22, 23]. Although our finding was inconsistent with Exelmans and Van den Bulck (2021) [9] and Kroese et al. (2016) [23], it was corroborated by the findings of Kühnel et al. [25], where trait self-control was unrelated to bedtime procrastination after controlling for chronotype and state self-control. A possible explanation for this result was that the present study did not differentiate procrastinatory media use from recreational media use, as the two forms of media use are driven by different psychological mechanisms: "recreational media use represents strategic delay, whereas procrastinatory media use represents irrational delay" (p. 5) [49]. In Reinecke and Hofmann's (2016) study, trait self-control significantly predicted procrastinatory media use but not recreational media use [49]. Therefore, the null effect of self-control on bedtime media use observed here might reflect the recreational nature of media use in the present study. Future research is needed to further test if recreational and procrastinatory media use would differently explain the relationship between chronotype and sleep quality.

Findings of additional analyses indicated that the mediation path from morningness to sleep quality through pre-sleep stress management was significant only for individuals high in trait self-control. For those low in trait self-control, morningness was unrelated to better stress management habits at bedtime. This suggests that individuals low in trait self-control are at a high risk of sleep problems as a result of inadequate sleep hygiene related to emotion regulation. Additionally, the finding that the mediation effect of pre-sleep stress management emerged only for individuals high in trait self-control indicated that only among these individuals are the benefits of being morning types transmitted to proper sleep hygiene habits and good sleep quality. Future work may clarity this process by, for instance, examining other individual characteristics that also influence emotional stability (e.g., neuroticism). Furthermore, the finding that chronotype and sleep hygiene habits jointly predicted sleep quality suggests that although eveningness has a negative impact on sleep quality, no such effect was found among evening types who already developed good pre-sleep stress management. This result lends support to the importance of adaptive sleep hygiene habits in regulating sleep among evening types.

## Implications for sleep hygiene recommendations

The present study pinpoints sleep hygiene as an underlying mechanism that explained the difference in sleep quality between individuals varying in trait self-control and chronotype. Our findings provide empirical evidence that morningness and high trait self-control were associated with better sleep quality through increased adaptive sleep hygiene habits, which presumably enable individuals to effortlessly engage in positive routinized practices for consistent and sufficient sleep. As adaptive habits are an effective means to promote healthy behaviors and prevent problematic regulation [12], interventions should prompt individuals to develop and regularly perform new routines for shaping pre-sleep behaviors.

The findings of the present study can guide the development of sleep hygiene recommendations in line with sleep characteristics associated with these individual differences. One of the

main findings implicates that pre-sleep stress management is a common factor that drives poor sleep quality in both evening types and those low in self-control; therefore, interventions incorporating mindfulness-based stress reduction appear to be promising to improve sleep quality in these individuals [50]. In addition to mindfulness training, other self-regulation techniques that have been shown to improve sleep quality include implementation intentions [18]. For example, a handful of studies with randomized controlled trials have demonstrated implementation intentions, which detail an "if-then" plan that guides people to enact a targeted behavior (e.g., "if it is 10 o'clock, I will get ready for bed"), to be effective to facilitate effortless goal-directed responses and thus reduce bedtime procrastination [18, 51]. Furthermore, as we have identified bedtime media use and sleep timing irregularity as highly relevant to poor sleep quality in evening types, whereas environmental interference (e.g., noise or disorganization) is linked to poor sleep quality in individuals low in self-control, these findings can inform the design and implementation of personalized sleep hygiene practices appropriate for the target population [17]. Taken together, this study suggests that more attention should be dedicated to understanding individual differences in sleep characteristics and relevant sleep hygiene habits, as well as to developing guidelines to assist individuals in identifying problems and treatments in accordance with their sleep characteristics.

### Limitations and future research

This study was the first to examine the multiple mediation of sleep hygiene behaviors in the relations between chronotype, trait self-control and sleep quality. However, there are several limitations to this study. First, given the correlational and cross-sectional nature of this study, future replication with pre-registered studies is necessary to establish the robustness of the current findings. Furthermore, conceptual replications with different operationalizations or more rigorous methods (e.g., longitudinal design) are particularly important for advancing the understanding of how individual differences in circadian rhythms and trait self-control determine sleep hygiene behaviors and influence sleep quality. As past research suggests that poor sleep hygiene and delay of circadian rhythm tend to reinforce each other [34], replicating the current findings with a longitudinal study and cross-lagged panel analyses would be of significant value for illuminating the reciprocal and dynamic associations between chronotype, trait self-control, sleep hygiene, and sleep quality. Second, the convenience sampling method of this study resulted in a skewed age- and gender-ratio, which limit the generalizability of the findings. Future research is needed to test the mediation pathways in a more nationally representative sample in Hong Kong. Third, this study relied on self-report, and thus the data may have been subjective to recall and measurement biases [52]. Although a meta-analytic report has indicated that sleep behaviors can be studied reliably with self-report data as compared to sleep diaries [53], there may be a gap between self-reported and actual behaviors. Future studies should include objective assessments of sleep quality (e.g., actigraphy). Fourth, despite controlling for demographic characteristics as covariates (i.e., age, gender and marital status), other potential covariates that were not controlled for include shared bedroom and the presence of sleep problems (such as sleep apnea or insomnia). Future studies could benefit from including these covariates known to affect sleep quality.

In conclusion, the current investigation provided evidence that chronotype and trait self-control distinctively predicted sleep quality in Chinese adults in Hong Kong. Sleep hygiene was found to be significant mediators, in parallel with bedtime media use, of the associations between chronotype, trait self-control and sleep quality. These results supported the protective effect of sleep hygiene as adaptive habits that enable individuals to be less susceptible to poor sleep quality. Furthermore, as sleep hygiene represents behavioral factors of sleep that can be

ameliorated by goal-directed action, this study provides a rationale for improving sleep quality in evening types and people low in trait self-control by making them become aware of and purposefully adopt good sleep hygiene practices.

## Supporting information

**S1 Table. Results from moderated mediation model.**
(PDF)

**S1 Fig.**
(TIF)

**S2 Fig.**
(TIF)

## Author Contributions

**Conceptualization:** Shiang-Yi Lin, Kevin Kien Hoa Chung.

**Data curation:** Shiang-Yi Lin.

**Formal analysis:** Shiang-Yi Lin.

**Funding acquisition:** Kevin Kien Hoa Chung.

**Investigation:** Shiang-Yi Lin.

**Methodology:** Shiang-Yi Lin, Kevin Kien Hoa Chung.

**Project administration:** Kevin Kien Hoa Chung.

**Resources:** Kevin Kien Hoa Chung.

**Software:** Shiang-Yi Lin.

**Supervision:** Kevin Kien Hoa Chung.

**Visualization:** Shiang-Yi Lin.

**Writing – original draft:** Shiang-Yi Lin.

**Writing – review & editing:** Shiang-Yi Lin, Kevin Kien Hoa Chung.

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
