## [Decision Letter · Decision Letter 0]

4 Nov 2021

PONE-D-21-19877Chronotype and trait self-control distinctively predicted subjective sleep quality via sleep hygiene behaviors and bedtime media usePLOS ONE

Dear Dr. Chung,

Thank you for submitting your manuscript to PLOS ONE. After careful consideration, we feel that it has merit but does not fully meet PLOS ONE’s publication criteria as it currently stands. Therefore, we invite you to submit a revised version of the manuscript that addresses the points raised during the review process.

I was fortunate enough to received two very detailed and thoughtful reviews from experts in the field regarding the suitability of your manuscript for publication. Both reviewers felt that your paper was well-written and addressed important issues with respect to the consequences of self-control and chronotypes. However, they also both had some serious concerns about your use of the strength model of self-control, both in terms of the controversy surrounding this model, and also its appropriateness for justifying examining individual differences in self-control. As Reviewer 2 aptly points out, that model describes intra-personal variances in self-control, not differences between individuals which is what is captured when self-control is measured with a trait measure. Accordingly, the conclusions drawn with respect to this model are not well founded or theoretically logical. I agree with both the Reviewers that this model should be replaced with a more appropriate model for describing individual differences in self-control. The Reviewers also raised several concerns about the methods and analyses, and how they were presented. After reading their comments and the manuscript myself, I too was left wondering why age had been dichotomised instead of keeping it as a continuous variable in the analyses. There doesn’t appear to be any strong justification for treating this variable this way, and I would suggest that in the absence of any compelling rationale, that the analyses be rerun leaving age as a continuous variable. These are just two of the key concerns that need to be addressed in a revision. The reviewers have raised other concerns as well that require serious consideration when revising your manuscript.

We look forward to receiving your revised manuscript.

Kind regards,

Fuschia M. Sirois, PhD

Academic Editor

PLOS ONE

Journal Requirements:

Reviewers' comments:

Reviewer's Responses to Questions

**Comments to the Author**

1. Is the manuscript technically sound, and do the data support the conclusions?

Reviewer #1: Partly

Reviewer #2: Partly

2. Has the statistical analysis been performed appropriately and rigorously? 

Reviewer #1: Yes

Reviewer #2: No

3. Have the authors made all data underlying the findings in their manuscript fully available?

Reviewer #1: Yes

Reviewer #2: Yes

4. Is the manuscript presented in an intelligible fashion and written in standard English?

Reviewer #1: Yes

Reviewer #2: Yes

5. Review Comments to the Author

Reviewer #1: PLOS One – Chronotype, Self-Control, and Sleep Quality

1. ADEQUACY OF LITERATURE REVIEW:

a. This manuscript includes much of the relevant literature on the topic, I found it easy to follow and read.

b. Given the now highly notable failures to replicate, I don’t believe the Strength Model of self-regulation is an appropriate theoretical framing of these findings. That being, I don’t believe this theory actually leads to any specific predictions in this paper and I would recommend another dominant model of self-regulation to situate these predictions.

2. THEORY TESTING:

a. Outside of the issues with using the strength model of self-regulation, I thought the predictions were clearly written and fairly intuitive to follow.

b. The authors state “The relevance of chronotype for sleep hygiene is apparent” but I think this point is not clearly apparent and I would prefer to see a bit more elaboration on how/why there is a link between hygiene and chronotype

3. KNOWLEDGE GAINED:

a. Overall, I thought the manuscript was well-written, clearly situated within the literature, and on an important topic that we stand to gain much by understanding further.

4. METHODOLOGY:

a. The biggest issue I had with this manuscript is its primary reliance on using cross-sectional, correlational data to analyze the multiple mediation model. I think for this project to represent a meaningful contribution to the literature there needs to be a second study (ideally some kind of experimental work or, perhaps more practically, a longitudinal study, or a case-control approach) with some other, stronger form of evidence. As it stands now, these analyses merely represent a series of concurrent correlational associations already found in the literature. Without stronger forms of evidence, I think the ultimate contribution of this paper is limited. Alternatively, a pre-registered replication might alleviate at least the concern of this pattern of results’ stability.

b. Were there any exclusion criteria? For example, individuals working the night shift.

5. STATISTICAL ANALYSES:

a. I think, generally speaking, the analyses are well-supported by the literature review.

b. The proposed multiple mediation model section (p. 7, lines 139 – 145) should be placed in its own section following the methods section, prior to the results. Something akin to an Analytic Plan, containing other relevant analytic details to orient readers to the results section.

c. Why were the number of sleep hygiene components derived from an exploratory factor analysis and not from prior psychometric work on the Sleep Hygiene Index.

d. Why was age discretized at age 40? It seems more advantageous to leave this in your models as continuous (p. 15, line 287)

e. Were the results substantially different when the covariates were omitted from the model?

6. INTERPRETATION:

a. I thought the interpretation of the results were clear and well-situated within the larger literature in the discussion section.

b. I would like to see a more involved discussion of the univariate and multivariate descriptives statistics in this sample. I think readers don’t get a very good sense of who this sample really is – are they a more evening oriented group? Is there self-control better than average? As it stands now, the descriptive/preliminary analyses (p. 13, lines 255 – 264) skews heavily towards describing the sleep behaviors and other media activities, more could be gleaned from other descriptives.

7. LIMITATIONS:

a. With 83.5% of the sample comprising women, I wonder the extent to which this sample can reasonably generalized to men.

b. Were moderation analyses conducted instead of simply looking at covariates?

c. The authors observed fairly low reliability for the rMEQ—I wonder if they could speculate about why this might be the case, as well as if there are any potential solutions for improving reported reliability (e.g. is there an item that seems to be acting up in this sample?).

8. FUTURE DIRECTIONS/APPLICATIONS:

a. I think it would be nice to see a richer discussion of how these results might directly facilitate intervention development.

b. Specifically, other work on bedtime procrastination and self-regulation interventions (e.g. Valshtein et al., 2020) and other interventions could be integrated into the final discussion paragraph

Reviewer #2: The authors analyze the influence of chronotype and differences in self-control on the subjective assessment of sleep quality. They assume that the relationship between chronotype, self-control, and sleep quality is mediated by sleep hygiene practices and the bedtime media use.

Below, I make comments and remarks on the following parts of the manuscript.

Abstract:

1) The statement (p. 2) "Chronotype and trait self-control positively predicted sleep quality" is imprecise and its correct interpretation requires the knowledge that the higher the score on the chronotype measuring scale, the more morning type the person is. The wording "late chronotype" is likewise imprecise.

Introduction:

1) It is not clear to me why the authors start from the self-control strength model and then refer to this model several times since they analyze individual differences in self-control? After all, they do not analyze self-control as a state, but as a trait.

2) The authors wrote that „their study focuses on two individual differences that determine self-regulation abilities - i.e. chronotype and trait self- control ” (p.3). This sentence requires clarification, in what sense does the chronotype determine the ability to self-regulate? The authors mean the general ability to self-regulate, or the ability to self-regulate sleep behavior?

3) Again, the authors describe the self-control strength model and then state that the self-control trait is a stable individual difference (p.3). In the self-control strength model, intra-individual differences in self-control are analyzed, not differences between individuals.

4) The authors repeated that "another characteristic that determines self -regulation abilities and contributes to sleep quality is chronotype." (p.5) Again, are the authors concerned with general self-regulation abilities or just regulation of sleep behavior?

5) In the part describing the relationship between the chronotype and sleep hygiene and sleep quality, the authors mention the relationship between eveningness and procrastination, often confirmed in research (p. 6). However, the authors do not mention that the research also indicates a relationship between chronotype and self-control trait. Thus on the basis of the research conducted so far, the assumption made by the authors about the independent effect of self-control and chronotype seems unjustified. While it is still unclear about the direction of the relationship between self-control and chronotype, it seems obvious that the two features are related. It also means that the model of analysis adopted by the authors in the following part is not justified and raises significant doubts.

Statistical analysis:

1) Why do the authors provide age ranges, but mean value, standard deviation, and age range are missing? Why do the authors divide age into two categories under 40 and over 41 instead of including age as a continuous variable? Age is an important factor influencing the chronotype and quality of sleep. With this approach to this variable, the relationship between age and chronotype and sleep quality is impossible to capture.

2) The results of the correlation analysis confirm that self-control and chronotype are significantly (p <.001) related. In my opinion, this relationship does not allow these variables to be included in the analysis as two distinctive independent variables.

3) Further analyzes confirm that the effect of self-control on sleep quality is bigger than that of the chronotype and that it is both direct and indirect.

Discussion:

1) The authors wrote: :” In accordance with the self-control strength model, the abilities to downregulate stress and emotion (e.g. self-regulatory abilities) underlie the differences in sleep quality between morning and evening types and between people high vs. low in trait self control”. (p.20) - I do not understand in what sense it is consistent with the self-control strength model? In addition, it is about abilities or some habitual behaviors? Sleep hygiene is associated with certain healthy habits, not abilities.

2) The discussion does not explain the lack of influence on sleep quality in the studied group of such sleep hygiene behaviors as sleep timing regularity and avoiding sleep-disrupting food or activities. Why did these behaviors, known to positively affect sleep quality, not have such an impact on sleep quality in the study group?

3) Pointing to the limitations of the conducted study, the authors emphasize that sleep habits should be measured in a different way in the future. While in the case of sleep habits self-reports methods seem justified, when assessing sleep quality one should go beyond self-reports.

4) In the future, it would also be worth ensuring a larger and more representative group of participants.

5) In conclusion, the authors state that they provided evidence that „chronotype and trait self-control distinctively predicted sleep quality in Chinese adult in Hong Kong”. In my opinion self – control and chronotype are related features and therefore cannot be considered as „distinctively predicted sleep quality”. In addition, the results of the study of a non-representative group of 224 participants (predominantly women, 83.5%) do not provide a basis for generalizing conclusions about the influence of chronotype and self-control on sleep quality in „Chinese adult in Hong Kong ”.

The authors analyze the socially important, extremely current problem of low-quality sleep and try to explain the causes of sleep problems by pointing to the role of features such as chronotype and self-control in regulating sleep-related behavior. The authors also try to point out the mechanism of the influence of these features related to sleep hygiene and bedtime media use. In my opinion, the authors only partially achieved the intended goal, for the reasons indicated above. On the positive side of the work, I would include the analysis of several different sleep hygiene behaviors and the construction of their own method - the Bedtime media use analysis indicator.

6. PLOS authors have the option to publish the peer review history of their article (what does this mean?). If published, this will include your full peer review and any attached files.

Reviewer #1: **Yes: **Timothy J. Valshtein

Reviewer #2: No

---

## [Author Response · Author response to Decision Letter 0]

20 Jan 2022

The response letter has been uploaded as a word document.

---

## [Decision Letter · Decision Letter 1]

30 Mar 2022

Chronotype and trait self-control as unique predictors of sleep quality in Chinese adults: the mediating effects of sleep hygiene and bedtime media use

PONE-D-21-19877R1

Dear Dr. Chung,

We’re pleased to inform you that your manuscript has been judged scientifically suitable for publication and will be formally accepted for publication once it meets all outstanding technical requirements.

Kind regards,

Fuschia M. Sirois, PhD

Academic Editor

PLOS ONE

Reviewer's Responses to Questions

**Comments to the Author**

1. If the authors have adequately addressed your comments raised in a previous round of review and you feel that this manuscript is now acceptable for publication, you may indicate that here to bypass the “Comments to the Author” section, enter your conflict of interest statement in the “Confidential to Editor” section, and submit your "Accept" recommendation.

Reviewer #1: All comments have been addressed

Reviewer #2: (No Response)

2. Is the manuscript technically sound, and do the data support the conclusions?

Reviewer #1: Yes

Reviewer #2: Yes

3. Has the statistical analysis been performed appropriately and rigorously? 

Reviewer #1: Yes

Reviewer #2: Yes

4. Have the authors made all data underlying the findings in their manuscript fully available?

Reviewer #1: Yes

Reviewer #2: Yes

5. Is the manuscript presented in an intelligible fashion and written in standard English?

Reviewer #1: Yes

Reviewer #2: Yes

6. Review Comments to the Author

Reviewer #1: After viewing the responses to the comments I and the other reviewer have laid out, I feel that this manuscript is suitable for publication. I still have some reservations about the ultimate contribution of a cross-sectional mediation analysis, but the manuscript's clear writing, straightforward hypotheses, and non-WEIRD sample are sufficiently meaningful contributions to merit publication in PLOS one. I commend the authors on a thoughtfully revised manuscript.

Reviewer #2: I would like to thank the authors for responding to all my comments. In my opinion, the Authors, following the comments of all reviewers, significantly improved their manuscript. It is worth noting not only a significant improvement in the introduction, but also in statistical analyzes, including conducting additional analyzes - moderated mediation. Consequently, the authors managed to explain the obtained results much better.

The method of measuring and analyzing the age of the respondents remains a weakness of the study. I also suggest that the results of the additional mediation analysis be included not in the discussion (p.27) but in the Results section.

I recommend the manuscript for publication.

7. PLOS authors have the option to publish the peer review history of their article (what does this mean?). If published, this will include your full peer review and any attached files.

Reviewer #1: **Yes: **Timothy J. Valshtein

Reviewer #2: **Yes: **Romana Kadzikowska-Wrzosek

---

## [Editor Report · Acceptance letter]

4 Apr 2022

PONE-D-21-19877R1 

Chronotype and trait self-control as unique predictors of sleep quality in Chinese adults: the mediating effects of sleep hygiene habits and bedtime media use 

Dear Dr. Chung:

I'm pleased to inform you that your manuscript has been deemed suitable for publication in PLOS ONE. Congratulations! Your manuscript is now with our production department. 

Kind regards, 

on behalf of

Dr. Fuschia M. Sirois 

Academic Editor

PLOS ONE